# CanCOG^®^: Cultural Adaptation of the Evidence-Based UCLA Cognitive Rehabilitation Intervention Program for Cancer Survivors in Portugal

**DOI:** 10.3390/healthcare11010141

**Published:** 2023-01-02

**Authors:** Ana F. Oliveira, Milaydis Sosa-Napolskij, Ana Torres, Diâner Felipe Queiroz, Ana Bártolo, Helena Sousa, Sara Monteiro, Kathleen Van Dyk, Linda M. Ercoli, Isabel M. Santos

**Affiliations:** 1Center for Health Technology and Services Research of the Health Research Network (CINTESIS@RISE), Department of Education and Psychology, University of Aveiro, 3810-193 Aveiro, Portugal; 2Center for Health Technology and Services Research of the Health Research Network (CINTESIS@RISE), Faculty of Medicine, University of Porto, 4200-319 Porto, Portugal; 3Department of Psychology and Education, Faculty of Human and Social Sciences, University of Beira Interior, 6200-209 Covilhã, Portugal; 4Department of Education and Psychology, University of Aveiro, 3810-193 Aveiro, Portugal; 5Center for Health Technology and Services Research of the Health Research Network (CINTESIS@RISE), Piaget Institute—ISEIT/Viseu, 3515-776 Viseu, Portugal; 6Department of Social Sciences and Management, Open University, 1269-001 Lisboa, Portugal; 7Department of Psychiatry & Biobehavioral Sciences, Jane & Terry Semel Institute for Neuroscience & Human Behavior, University of California Los Angeles, Los Angeles, CA 90024, USA; 8Jonsson Comprehensive Cancer Center, University of California Los Angeles, Los Angeles, CA 90024, USA; 9William James Center for Research (WJCR), Department of Education and Psychology, University of Aveiro, 3810-193 Aveiro, Portugal

**Keywords:** cancer, cancer survivors, cancer-related cognitive impairment, cognitive rehabilitation, cultural adaptation

## Abstract

Cognitive difficulties are highly prevalent and negatively impact cancer survivors’ quality of life. The UCLA Cognitive Rehabilitation Intervention Program (in short, UCLA program) is an evidence-based intervention developed and tested in the US to address the cognitive complaints of cancer survivors. Since there are no cognitive rehabilitation programs available for Portuguese cancer-related settings, this study aimed to culturally adapt the UCLA program to Portugal. Nine steps were implemented for this cultural adaptation: needs assessment, initial contacts, translation, cultural adaptation, independent review by a panel of experts (n = 6), focus group discussions with cancer survivors (n = 11), systematization of inputs and improvement of the final materials, fidelity check, and preliminary acceptability assessment. The findings suggested that changes to the original materials were needed. A Portuguese name, “CanCOG^®^—Reabilitação Cognitiva no Cancro” (in English “CanCOG^®^—Cognitive Rehabilitation in Cancer”), and a logo were created to make it more memorable and appealing for the Portuguese population. The language was adjusted to ensure content accessibility and semantic and conceptual equivalence. Finally, references to several cultural aspects, such as habits, customs, and traditions, were adapted to fit the new cultural context. The UCLA program may be a promising tool to help alleviate the cognitive difficulties reported by cancer survivors in different cultural contexts. Future research is needed to confirm the feasibility, acceptability, and preliminary efficacy of its Portuguese version, “CanCOG^®^—Reabilitação Cognitiva no Cancro”.

## 1. Introduction

Cognitive difficulties are one of the most feared problems reported by patients treated for non-central nervous system (CNS) cancers throughout the disease trajectory [1,2,3,4]. This phenomenon, commonly referred to as cancer-related cognitive impairment (CRCI) and colloquially known as *chemobrain*, can be described as cognitive symptoms reported by cancer patients in self-reported questionnaires or as cognitive changes evaluated by formal neuropsychological tests in several cognitive domains [5,6], such as executive functions, language, verbal and working memory, processing speed, motor function, and attention and concentration [7]. The reported prevalence of CRCI for non-CNS cancers is somewhat variable [1,4,5,6,8], but evidence suggests that it is highly prevalent in Europe (40–75%) [9,10]. Even though data from the Portuguese population is still scarce, a study performed in Northern Portugal found that the incidence of cognitive impairment one year after diagnosis was estimated to be 8.1% in women with breast cancer [11].

Compared to degenerative conditions and other CNS disorders, CRCI is generally subtle [3,12]. Nevertheless, it can significantly affect the quality of life (QoL) and the ability to function in different aspects of cancer survivors’ life, such as occupational, social, and daily life activities [3,6,13,14,15]. Considering the negative impact of CRCI on QoL in the context of long-term cancer care, cancer survivors have a growing need and demand to implement interventions to manage CRCI [9]. A large survey conducted in France demonstrated that 75% of participants (>85% breast cancer survivors) wished to receive support for CRCI, particularly cognitive training (72%) [16]. This data emphasizes the importance of monitoring CRCI and identifying effective and accessible intervention options for CRCI to allow cancer survivors to return to everyday life and improve their QoL [16,17,18,19].

Literature aiming at examining the efficacy of evidence-based interventions to manage CRCI is growing slowly, remaining a clinical challenge [5,20]. Various interventions have been developed and tested to treat CRCI, including pharmacological and non-pharmacological approaches [5,6,13,21]. Non-pharmacological treatments, including cognitive training and rehabilitation, exercise, and mind-body interventions, have received the most support from empirical research [5,13,15,18,19,21]. Specifically, National Comprehensive Cancer Network Survivorship Guidelines identify cognitive rehabilitation as a first-line intervention to address CRCI [18,22]. Cognitive rehabilitation is defined as systematic, functional, and behavioral-oriented interventions whereby people with brain injury work together with healthcare professionals and others to improve cognitive performance and remediate, alleviate, or manage/compensate cognitive deficits arising from a neurological injury, through the identification of personally relevant goals [3,6,23,24]. Therefore, these programs encompass the development of individualized skills to support cognitive deficits, assist with problem-solving, and improve or restore functioning in everyday tasks [2,5]. People either: (1) perform specific cognitive training, which involves the regular practice of skills in an attempt to restore cognitive functions; (2) learn how to manage cognitive deficits, for instance, to avoid or decrease distractions, plan and organize time, utilize strategies as calendars and mnemonics; and/or (3) receive education about brain functioning, cognitive deficits, and their implications for the instrumental activities of daily living [3].

Recent reviews provide promising evidence of cognitive rehabilitation being feasible to deliver, satisfactory to participants and positively affecting objective (performance on neurocognitive tests) and subjective (self-reported cognitive function) cognitive functioning in non-CNS cancer survivors. Nevertheless, generalization to daily life and other aspects of functioning (e.g., mood, fatigue) are understudied [2,17,18,19]. Regarding performance measures, improved memory has been most observed, followed by executive functions, processing speed and visuospatial skills, with small to large intervention effect sizes. Patients also report fewer cognitive complaints or improved cognitive abilities [2,18]. These interventions are delivered on either an in-person manualized or home-based (e.g., videoconference, telephone) individual or group format, over multiple sessions (ranging from 4–12 weekly or biweekly sessions, from 20–120 min each), with or without homework, and telephone support between sessions [2,5,18]. Additionally, computer and web-based programs are being increasingly evaluated [3,25].

Regarding the offer of programs that target CRCI in Portugal, there is still no intervention specifically developed for this purpose. To the best of the authors’ knowledge, one online cognitive training program was used in this context with brain tumor patients [26]; however, it lacks empirical validation of its effectiveness with the cancer population as it was developed for neurodegenerative diseases [1]. This landscape stresses the need to develop specific programs targeted at cancer survivors with cognitive difficulties in Portugal. Among the international programs already created and with evidence to improve CRCI, the UCLA Cognitive Rehabilitation Intervention Program (hereafter referred to as UCLA program) was specifically developed for cancer survivors with cognitive complaints [27,28]. The development and pilot-testing of this program was started in 2008 by a team led by Dr Linda M. Ercoli at the University of California, Los Angeles (UCLA), in the US [27]. The program was originally developed to use with breast cancer survivors [27], but further work has been carried out to adapt it to all types of cancer and to disseminate it as a licensed intervention for hospitals in the US. The 5-week manualized intervention program is rooted in evidence-based cognitive rehabilitation and targets the most common complaints of cancer survivors: attention, executive functions, and memory (see Figure 1). The intervention is described in detail in prior publications [27,28], but it is also presented in Table 1, following the “TIDieR checklist (Template for Intervention Description and Replication)” [29].

In recent years, a growing interest in CRCI has emerged in Portugal [30,31,32,33,34,35]. However, the work developed has mostly characterized cognitive complaints and deficits without focusing on developing and implementing cognitive interventions. The UCLA program was developed and tested for the US population, thus requiring translation and cultural adaptation if one intends to use it in a different context. Therefore, given the absence of evidence-based cognitive rehabilitation programs for cancer survivors in Portugal, the aim of this study is to culturally adapt this intervention to the Portuguese context, thus assuring intervention salience and fit with the new context [36]. This process is the first experience of formally adapting this program to an international population and setting. There is evidence that culturally adapted interventions are more effective than interventions that have not been adapted [37,38,39] and that a culturally adapted intervention is similarly effective to its original version [40]. Implementing evidence-based interventions (i.e., those that have already been assessed to be effective) in new contexts may save financial, human, and time resources compared to developing new interventions for each context [41]. In this scope, cultural adaptations are appropriate and needed when there is a perceived mismatch between the program and the cultural context in which it is planned to be implemented [42], to accommodate different contextual features and resources [41] to achieve a better fit with the new context [43].

Cultural adaptation is the systematic modification or adjustment of culturally sensitive aspects of an evidence-based intervention, i.e., language, culture, and context, to match the target population’s cultural patterns, worldviews, meanings, and values [44,45,46]. These adaptations aim to maintain the fidelity of the core elements of the original intervention (i.e., components that are critical features of an intervention’s purpose and design and are sought to be responsible for its effectiveness [47]), while also adding some cultural content to the intervention and/or its methods for engaging participants [46]. Therefore, cultural adaptation intends to preserve the cultural relevance of an intervention when delivering it to different population groups [48] and involves a planned, organized, iterative, and collaborative process that habitually includes the participation of the target population for whom the adaptation is being developed [40]. Cultural adaptation can encompass adaptations to the surface structure, i.e., materials or activities that address observable and superficial aspects of a target population’s culture (e.g., people, language, places, product brands, music, foods, locations, clothing, and related observable aspects) and to the deep structure of a program, addressing the core values, beliefs, norms, customs, traditions, and lifestyles of the cultural target population, and requires an understanding of specific cultural nuances and cultural competence of the cultural adaptation team [40,49]. As Lee et al. [42] state, the heart of cultural adaptation is the balance between fidelity (i.e., accurate implementation of a program as conceived by the program developers) and cultural/ecological validity or fit (i.e., the extent to which a program is generalizable beyond the initial target population to other populations and cultural contexts) [50], resolving any mismatches and enhancing overall fit between program and context [42]. Thus, the ultimate goals of a cultural adaptation are to increase participants’ engagement (reflected in better recruitment and retention) and to enhance the impact on desired outcomes without compromising the core principles and effectiveness of the program [42].

This paper describes the development of a culturally adapted version of the UCLA program to the Portuguese context and the adaptation process of the program contents.

## 2. Materials and Methods

Cultural adaptation activities of the UCLA program to Portugal took place from August 2019 to January 2022, within the scope of the first author’s doctoral scholarship (A.F.O.). The project’s team comprised senior researchers, clinicians with expertise in psycho-oncology and neuropsychology/cognitive rehabilitation, and a trained and certified translator. This study was approved by the Ethics and Deontology Committee of the University of Aveiro, Portugal (22 January 2020/No. 30/2019).

### 2.1. Framework for the Cultural Adaptation Process

There is no single, correct, golden standard to culturally adapt interventions [51,52] nor consensus on the steps that need to be followed [53]. Over the past decade, various reviews have proposed theoretical approaches for cultural adaptation [36,40,43,54]. Two sets of cultural adaptation frameworks can be considered [51,54], which can be complementary [40]: (1) frameworks that inform the modification of the content of the intervention, i.e., “what” to adapt—for instance, the ecological validity model, which specifies eight domains corresponding to culturally sensitive elements to be adapted (the *language* of the intervention; *persons*, i.e., similarities and differences between the client and therapist; *metaphors*, i.e., cultural expressions and sayings that might be used in treatment; *content* or cultural knowledge; treatment *concepts*; *goals*; treatment *methods*; and *context* of the treatment) [44], and the cultural sensitivity model, which identifies adaptations to the surface versus deep structure [49]; and (2) frameworks that inform the process of adaptation, i.e., focus on the process or steps in “how” to conduct an adaptation, including decisions about when to adapt, how to adapt, and which stakeholders should be involved in the process [51]. The latter are known as comprehensive multistep frameworks or stage models. They contain a series of a priori deliberate steps that guide researchers in using qualitative and quantitative data to determine the need for a cultural adaptation, the elements of the intervention that need changes, and assessments of the effects of intervention alterations [40,54].

The UCLA program was culturally adapted to improve relevance, reach, and engagement with cancer survivors speaking Portuguese and living in Portugal. The process was based on international guidelines for the cultural adaptation of evidence-based interventions, namely public health interventions [36,43,55,56] and psychological interventions [42,47,52,57,58,59,60,61], including strategies from both bottom-up (i.e., contextual and cultural understanding shared by in-group members) and top-down (i.e., investigator-driven research guided by scholarship and theory) approaches to reach a balance between sensitivity and responsiveness to population needs and scientific integrity/fidelity [40,42,62]. Although guidelines for the cultural adaptation of cognitive interventions are available, namely cognitive stimulation interventions [63], these guidelines were insufficient to cover all the UCLA program components (i.e., culturally adapted translation of all materials, including the education and goal setting components). Therefore, a 9-step model was proposed, structured in three overarching phases (Figure 2), which are detailed in the subsequent sections

For this paper, all steps from the Exploration and Preparation Phases are reported. Preliminary acceptability results from the Implementation Phase are also described; full findings from this phase will be reported elsewhere.

The adaptation process described in the present work consists of important features that are mentioned in the literature [56]: involvement of stakeholders [36,64], i.e., the target population (cancer survivors) and experts (psychologists—effective members of the Portuguese Psychologists Association, Ordem dos Psicólogos Portugueses—with experience in research and clinical practice in the fields of psycho-oncology and neuropsychology); triangulates measures by using a mixed methods approach utilizing quantitative (i.e., independent review by a panel of experts) and qualitative (i.e., focus group discussions with cancer survivors and qualitative feedback on the program by a panel of experts) methods and data; and involves a certified translator with experience in translating health interventions.

Following the good practices of culturally adapting interventions, an adaptation plan [55,56] was developed to guide the research team on the program’s cultural adaptation process. An adaptation plan is crucial to document the process of adaptation of an intervention, addressing the questions of why the content and/or materials were adapted and what has been adapted. Thus, it provides a transparent and verifiable record of the adaptation process [56].

#### 2.1.1. Exploration Phase

**1. Needs assessment, identification, and review of evidence-based cognitive rehabilitation programs.** Initially, a needs assessment was conducted to determine the pertinence of adapting a cognitive rehabilitation program for cancer survivors to the Portuguese context [58]. This work included: (1) a *review* of empirical studies on the characteristics and cognitive intervention needs of cancer survivors; (2) an *analysis of databases* (e.g., Statistics Portugal, US Census) to compare the characteristics of Portuguese and American adult cancer survivors to identify similarities and differences between the two populations related to age and education level; and (3) a *collection of quantitative information* using a web-based survey to assess the access, needs and intervention preferences of Portuguese cancer survivors, to determine how well the intervention under study could correspond to their needs and preferences. The discussion of the findings on needs assessment will be reported elsewhere [65]. Furthermore, a *literature review of available evidence-based cognitive rehabilitation programs*, both in-person and web-based, was conducted. The search was performed by consulting published literature on non-pharmacological interventions for CRCI and, more specifically, cognitive rehabilitation programs [2,25,66,67,68,69,70,71,72].

**2. Selection of the intervention and initial contact with the original program development team to establish a collaboration and gather the original program materials.** After the literature review on the available programs was carried out, the UCLA program was selected for cultural adaptation [27,28]. This choice was based on its characteristics (short duration, focus on main cognitive complaints of cancer survivors, efficacy results) and the availability of the original program development team to collaborate. After contacting the author of the program (Dr Linda M. Ercoli) and requesting all the materials used in implementing the program, the resources were provided, and the members of the Portuguese research team kept in regular contact by e-mail to clarify doubts about the contents of the program. Additionally, a 2 h session was held, via Zoom, in September 2020, with three members of the Portuguese research team (A.F.O, A.T., and I.M.S.), about how to apply and use the program. Furthermore, the leading researcher of the project (A.F.O.) participated in a 5 h online training session, via Zoom, in October 2021 to observe memory training.

#### 2.1.2. Preparation Phase

**3. Translation of the program materials by a professional translator and technical accuracy check of the contents by the research team.** Translating the UCLA program from American English (EN-AM) into European Portuguese (PT-EU) took approximately eight months, from September 2020 to May 2021. The work was carried out by a professional translator (M.S.N.) working at the Faculty of Medicine of the University of Porto, who was a specialist in terminology and had experience translating intervention and education programs, instruments, and other health genres from the previously mentioned source and target languages [58]. From the linguistic perspective, the translation assumed a strategy based on the Skopos theory [73].

First, the leading researcher of the project (A.F.O.) and the translator met to discuss and agree on the procedures and guidelines to follow. It was established that the translation process would be iterative, phased, and continuous. Therefore, upon completing the translation of a document, the translator would send it to the leading researcher who, in turn, would forward it to a group of PT-EU native speakers of the research team (A.T. and I.M.S.) who were also experts in the field of cancer and neuropsychology, for feedback on the rendition into PT-EU in terms of idiomaticity, terminological and technical adequacy, and register. This scientific and technical accuracy check was performed by comparing the translated text to the original materials. The experts’ feedback would be informed to the translator, who would take the necessary steps to validate or adjust the translation. This procedure would be repeated until consensus was reached and a definitive version was produced. Lastly, it was decided that the first document to be translated should be the Training Manual as it contained all the information about the sessions and the methodological guidelines for the trainers. As such content was partly repeated in the other documents of the program, its rendition would facilitate the translation of the other materials. However, it was also decided that the translation of the other documents (Home Practice Exercises, Answer Keys, and the Class Handouts) should not be started until feedback from the experts about the translation of the Training Manual was given to avoid repeating verification and correction processes.

The translation process started with an initial evaluation of the material to verify the technical terminology and meaning in the field of cancer and cognitive rehabilitation. Terms and phrases were collected, and translation options were proposed and addressed with the leading researcher before the translation of the complete program which contributed greatly to the consistency and uniformity of the final product.

**4. Cultural adaptation of the program contents.** After translating the contents and its technical accuracy check, cultural incorporations were made into the program [49] from May to June 2021. The proposed adaptations were based on the cultural knowledge of three members of the research team (A.F.O, A.T., and I.M.S., all Portuguese natives) regarding the characteristics, experiences, norms, values, behaviour patterns, and beliefs of the Portuguese population. Their research and clinical practice expertise with cancer survivors and intervention programs were also considered. In addition, other professionals were consulted on specific content to reach a consensus (a psychologist with expertise in memory, a professor of Portuguese, a professor of Geology, and an archaeologist). The original program development team also had a crucial role during this step. Before conducting any changes to the original contents, specifically to the cognitive exercises, they were contacted to clarify their choices regarding what led them to choose certain contents for the exercises. Therefore, explanations were given, and guidelines were defined to follow the same thought process. All adaptations were reported for later fidelity checks by the original program development team. Thus, together with the translator and the original program development team, an adaptation plan was developed in which the program elements to be culturally adapted were identified:
Words and expressions appropriate to the target population, considering adjustments to the level of reading difficulty and education to ensure content accessibility.Words and expressions for semantic and conceptual equivalence.Personal names used throughout the program (e.g., exercises, daily examples).References to different cultural aspects, such as habits, customs and traditions.Instructions and/or content of some exercises without cultural correspondence to the Portuguese context. Regarding instructions, guidelines were provided by the original program development team to maintain the equivalence of the exercises (e.g., substitute words that appear in the text a similar number of times and consider the word classes—for instance, nouns are easier to follow than conjunctions, determinants, or pronouns).

It should be noted that some of these elements were identified by the translator during her work and were discussed during this step. It is also important to highlight that the core components of the program have not been changed (e.g., change in order, elimination or incorporation of exercises or themes). Additionally, other adaptations were made, discussed, and authorized by the original program development team, namely an update of some contents, the creation of a name and logo associated with the Portuguese version of the program, and the creation of a new cover and title page for the Portuguese version of the program materials. At the end of this step, the first draft version of the program (*Draft 1*) was available.

**5. Independent review of the translation and cultural adaptation of the Draft 1 version of the program by a panel of experts.** The *Draft 1* version of the translated and adapted content of the program was then independently reviewed by a panel of experts to collect feedback on the adequacy of the adaptation work carried out in the previous two steps. This review was carried out in June and July 2021. From this review, additional cultural adaptations were proposed, which were discussed in the following step with the target population to reach a consensus. This review was carried out independently by six national experts fluent in European Portuguese and English, all psychologists and researchers with diverse areas of expertise:
One assistant professor with previous research experience in psycho-oncology, including aspects related to reproductive and sexual health and psychosocial adaptation of young female breast cancer survivors (e.g., breast and gynaecological cancer) and clinical intervention in other chronic diseases (e.g., chronic kidney disease, chronic obstructive pulmonary disease, caregivers of people with dementia).One doctoral student in health psychology, having worked as a clinical and health psychologist in a cancer patient support association and at a national reference centre in oncology. She also had research experience in neurosciences, including the study of CRCI.One assistant professor with extensive clinical and research experience in psycho-oncology, including group intervention (cognitive–behavioural therapy and psychoeducation), psychosocial adjustment, needs, work ability, and reproductive concerns in young cancer survivors, focusing on both patients and their caregivers.One assistant professor with extensive research experience in neuropsychology, namely ageing, cognitive rehabilitation, cultural adaptation and validation of neuropsychological instruments (e.g., attention, executive functions).One full researcher with clinical and research experience in neuropsychology, namely cognitive neurosciences and cognitive assessment and rehabilitation, specifically in the field of executive functions.One full researcher with extensive expertise in human memory.

These experts were invited considering their profiles and background that matched the areas of expertise of the program contents. The first three experts reviewed the full program (i.e., the five sessions that integrate the program). The last three experts each reviewed one of the sessions of the program, taking into consideration their fields of interest: expert #4 reviewed weeks 1 and 2, expert #5 reviewed week 3, and expert #6 reviewed weeks 4 and 5 (this session was assigned to this expert considering its short length).

For this content review, experts received a “review guide” with instructions on the objectives and procedures to perform this task, adapted from [58]. Instructions differed if the experts reviewed the full program (i.e., the five sessions) or only one or two sessions. Experts were instructed to first critically and fully read the text of the program sessions and corresponding materials assigned to them (each session presents content in the Training Manual, Home Practice Exercises, Answer Keys, and the Class Handouts) and then conduct a ***qualitative appraisal*** of the translation and adaptation, making free comments and recommendations on the documents. The aim was not to propose deep modifications to the structure and contents of the program (e.g., add contents, sessions), but instead to appraise the translation and adaptation taking into account: (1) *Sensitivity* of the language used, promotion of the dignity of cancer survivors (e.g., presence of any stigmatising expression or term, with negative connotation); (2) *Clarity and comprehensibility* of the contents (e.g., sentence formulation), identification of “how Portuguese” the text felt when read and if the Portuguese language used corresponded to a standard accepted throughout the country (i.e., if it did not have words, expressions or constructions that were more common in a specific region of the country); (3) *Familiarity and accessibility* of expressions/terms/concepts to the target population (e.g., use of jargon); (4) *Theoretical, technical and cultural precision* of the concepts and terms (e.g., technical concepts used in the field of cancer and cognitive rehabilitation); (5) *Adequacy of the names and other cultural content* (e.g., traditions, practices, food, poems); and (6) *Adequacy of the cognitive exercises*, i.e., whether the respective guidelines given in the instructions are comprehensible, with a degree of difficulty considered appropriate for the population to which it is addressed and if the succession of exercises had a progressive/adjusted degree of difficulty. Other suggestions for improvement could also be made.

Additionally, they were required to perform a ***quantitative appraisal*** of the same parameters that were previously evaluated qualitatively for each session and/or for the global program (in the case of the experts that read the full program), considering the four distinct documents: each of the six parameters were evaluated on a scale of 1–4, which aimed to assess the need for revision based on whether the objective underlying each parameter had been adequately reached in that session (1 = fully review, 2 = needs major revision, 3 = needs minor revision, 4 = no revision needed). After receiving the reviews from all experts, three members of the research team (A.F.O, A.T., and I.M.S.) systematized and qualitatively analysed their comments, contributions, and recommendations. Disagreements regarding comments and recommendations from different experts regarding terms or expressions and content of cognitive exercises were later discussed with cancer survivors to reach a consensus based on the target population’s experiences and opinions.


**6. Appraisal of the content and cultural adaptation of the program by the target population (cancer survivors) through focus group discussions.**


The next stage involved the identification and focus group discussions with cancer survivors. The focus group discussions aimed to introduce the program to potential participants and discuss with them terms, expressions and other contents that raised disagreements between experts, contributing to a final consensus. Furthermore, the content of cognitive exercises, namely from weeks 3 and 4, were discussed, as these exercises comprised several texts with cultural content.

Cancer survivors were recruited to participate in one of two focus groups conducted via Zoom, in September 2021. In total, 11 participants were included in the focus groups, the first with six and the second with five elements, to maximize individual participation and minimize group inhibition [74]. Each session lasted approximately 120 min. Both female and male cancer survivors were invited to participate if they agreed to do so voluntarily, were 18 years or older, had been diagnosed with non-CNS cancer, had completed active cancer treatments for at least 6 months and were able to read and understand Portuguese. The recruitment was carried out using a snowball procedure [75], starting with participants from previous studies that agreed to be contacted for further investigations of the project. The participants were sent an e-mail with the invitation and information about the study. If they agreed to participate, a second e-mail would be sent with a link to an online informed consent and a brief sociodemographic and clinical questionnaire used to collect data on sex, age, education level, work status, cancer diagnosis, date of diagnosis, and past and current/future treatments. In addition, some exercises of week 3 that consisted of longer texts were sent to the individuals before their participation, explaining that it was important to read them before the interview, considering that they demanded a more careful reading and examination for further discussion. Two researchers (A.F.O. and A.B.) with previous experience conducting qualitative research led the focus group discussion. The focus groups agenda included: (a) welcome and presentation of the facilitators; (b) informing on the general aims of the task; (c) motivating for participation, assuring that there were no right or wrong answers; (d) confidentiality and participation agreement; (e) permission request to record the session; (f) presentation of the cancer survivors; (g) brief presentation of the program, to explain its structure and contents; (g) task instructions; and (e) final questions to assess the acceptability of the program. It was explained to the participants that the aim of the focus group was for them to share their opinions as key informants on the contents presented. The participants were informed that the program materials were translated by a professional translator and culturally adapted from the US to Portugal, and reviewed by field experts. However, some contents raised doubts to the research team and experts, and their feedback would be of utmost importance to reach a final consensus concerning the suitability of the adapted materials. They were finally told that they would be asked about the appropriateness, comprehensibility, familiarity, and accessibility of the contents in Portuguese culture and if they were relevant to the target population. Firstly, some technical (e.g., endocrine therapy versus hormonotherapy, multitasking) and language-related (e.g., incorrect answers versus wrong answers) concepts were discussed, followed by some sentences that experts were unsure if the target population would understand. Next, examples of all cognitive exercises that integrated the program were presented, and the participants were asked if the instructions were clear and the content was culturally appropriate. More suitable alternatives were requested in cases where those objectives were not achieved. The final part of the focus group included a set of questions to assess the program’s acceptability to potential participants. A semi-structured guide was adapted from [55] and intended to gather information about cancer survivor’s perspectives on the potential impacts and benefits of the intervention; potential facilitators and barriers to participation; and opinions regarding the contents, structure, and organization of the program, its format and alternative formats of delivery (see Appendix A, for information on the specific questions that were included in the interview guide).

The focus group discussions were visual and audio-recorded. Then, the audio was transcribed verbatim, analysed, and coded by two independent research team members (A.F.O. and A.B.). A thematic analysis was used following the five phases proposed by Braun and Clark [76]: (i) familiarizing with the data, (ii) generating initial codes, (iii) searching for themes, (iv) reviewing the themes, and (v) defining and naming themes. Codes were grouped into potential themes, gathering all relevant data based on the initial questions included in the focus group guide. The codification process was conducted using the software ATLAS.ti 22. The Consolidated Criteria for Reporting Qualitative Research (COREQ) checklist was followed to ensure transparency, rigor, and comprehensiveness on aspects of the research team, methods, context of the study, findings, analysis, and interpretations [77] (see Appendix A).

**7. Systematization of previous contributions and recommendations from experts and target population and improvement of the final text materials of the program.** After all the experts and target population had commented and made their contributions and recommendations, the research team (A.F.O., A.T., and I.M.S.) systematized and analysed that input, from a qualitative perspective. Adaptations to the program contents were implemented and the final text materials of the program were obtained, resulting in the *Draft 2* version. All changes were reported.

**8. Fidelity check by the original program development team reporting the adaptation proposals and respective justifications for later approval and obtaining the final Portuguese version of the program.** The adaptation proposals were presented to the original authors, with the respective justifications, for later approval to obtain the *Final Version* of the program. The original authors were asked to analyse the fidelity of the adaptations, verifying if these were in line with the objective and main concepts of the original version of the program. Although this last step refers to the approval of the final changes, it is worth noting that the Portuguese and the original program development teams were constantly in contact. Therefore, before introducing any major adaptation to the contents (e.g., substituting the text of an exercise), the original team provided insight into the choice of the original contents to help find an equivalent form for the Portuguese version.

## 3. Results

This section presents the results of the cultural adaptation of the UCLA program contents to Portugal. Although results from the needs assessment are discussed elsewhere, for this study, some preliminary data are presented in the Appendix A. This table highlights the main mismatches between the characteristics of the UCLA program participants and potential Portuguese cancer survivors that are important to consider for the present study findings.

### 3.1. Translation and Cultural Adaptation of the Program Contents

The first element of the program that was culturally adapted was its name. From the very beginning of the adaptation process, the research team discussed that the direct translation of the original name to PT-EU would not be enough for the program to be recognized by the Portuguese population. Therefore, after brainstorming, the research team proposed the name “CanCOG^®^—Cognitive Rehabilitation in Cancer”, in Portuguese “CanCOG^®^—Reabilitação Cognitiva no Cancro” (in short, CanCOG^®^). This name is composed of the first three letters of the main terms of the program, “Cancer” and “Cognitive”. Additionally, a logo was created to make the program more memorable and appealing (Figure 3). Concerning chromaticity, the research team wanted to choose colors related to movements associated with the fight against cancer. Therefore, the official colors of World Cancer Day were selected for the logo (i.e., orange and blue). Finally, a brain-like image was chosen to identify this program as one designed to rehabilitate the brain.

The process of culturally adapting the UCLA program started with analyzing the texts concerning the topicality of the CRCI-related contents. Since the contents were written in 2008, it was necessary to examine whether the data aligned with recent evidence. In this regard, no information was considered outdated.

The translation process started with an initial evaluation of the material, which yielded a database with 109 terms and phrases whose translation into PT-EU was proposed by the translator and cross-checked (i.e., confirmed, 66%; or rectified, 33,94%) by the leading researcher. The number of translated words was roughly 43,000, corresponding to the texts of the Training Manual, Home Practice Exercises, Answer Keys, and the Class Handouts. Most of these texts did not pose relevant translation problems, and their rendering into PT-EU was straightforward. Nevertheless, minor adjustments were needed [58], as described next.

Considering that a high number of Portuguese cancer survivors are 65 years old or older and that they have low education levels (i.e., equal to or less than 4 years of education) [78], some *adjustments to reading and education levels* were considered. Therefore, some words and expressions were translated and adapted to be more appropriate to the target population to ensure content accessibility. The use of technical terminology was maintained, but an informal and straightforward verbal style was chosen. The inflection of words for the masculine gender, which is normative in standard Portuguese grammar, was used, and the form of addressing the reader using the second person of the singular, considered disrespectful in Portuguese, was changed to the third person (e.g., “sente-se distraído” and “sente-se frustrado”, from the English “feel distractible” and “feel frustrated”). In some cases, a more complex formulation of sentences was needed: for instance, in the original “Listening to a paragraph for a specific word”, in PT-EU, “Listening to the reading of a text and paying attention to a specific word”. Some long and complex sentences were split in two to simplify the text. This separation of sentences is a common procedure used in the translation of programs [58].

*Linguistic and conceptual equivalence* was also deemed necessary to ensure that the language used was adjusted so that it has meaning to the target population and that the same word or expressions define the same or analogous meaning to the target population, respectively [79]. In one example, in the original texts, the word “course” was used to refer to the program; however, its direct translation to PT-EU “curso” does not have the same meaning in Portuguese. Therefore, the word “programa” (from the English “program”) was adopted instead to refer to the context of the intervention. “Cognitive enhancement” was also used in the original texts to refer to the approach to enhancing cognitive functions. Nevertheless, in PT-EU, there is not a direct translation of this expression, so the translation of “Cognitive stimulation” was chosen.

Another example is the word “Testing” used in the context of the assessment time points of the intervention. Although the direct translation to PT-EU would be “Teste”, this word would not be well accepted by the target population since it has a connotation of testing knowledge (as happens at school). Thus, the expression “Momento de avaliação” (“Assessment moment”) was preferred. “In-class exercises” was also an expression that needed careful translation. “Exercícios para fazer na sessão” (Exercises to do during the session) was preferred to the more direct translation “Exercícios para fazer na aula” since the latter is related to the classes that students attend.

The original program contains auditory exercises. At the time of the development and implementation of the program, it was common to use CDs to listen to content. However, CDs are rarely used nowadays; therefore, it was proposed to give participants USB pens or send the audio files by e-mail, to match the current practices.

Throughout the texts of the program, specifically in examples and exercises, *personal names* were used. Eleven English names were substituted for Portuguese names. Whenever it was available, a commonly known, culturally accepted direct translation into Portuguese was used (i.e., Mary—Maria, Elena—Helena; Tanya—Tânia, Fred—Frederico). Despite having equivalents available in Portuguese (i.e., Roberto, Valéria, and Nicolau), the names Robert, Valerie, and Nick were adapted to Rui, Vânia, and Nuno for being more commonly used in Portugal while keeping the initials of the names in the original text. Furthermore, even though the name Steven has an equivalent name in Portuguese, i.e., Estêvão, it was preferred to use the name Miguel since it is a more common name in Portugal. Two of the names do not have any equivalent in Portuguese and therefore were substituted based on the similarity of spelling and pronunciation (Jamal—Jaime) and probably the most commonly known name in Portuguese (Hank—João, i.e., John). Finally, the only full name in the program (Fran Miles) was adapted to its equivalent in Portuguese, Francisco, plus the surname Miranda to keep the initial letter.

The most extensively adapted text of the program materials concerned the *content of some exercises* that included *references to different cultural aspects*, such as habits, customs, and traditions. Exercises could either be written or auditory and were managed separately. A significant part of an adaptation work refers to the cultural equivalence of the contents to ensure that the adapted program contains information that matches the local culture [58]. Therefore, it was necessary to find equivalents for several categories, such as food, animals, places, streets, and customary practices, based on the cultural knowledge of the research team and empirical studies, the contact with professionals from several areas, the suggestions made by the expert panel, and the opinions of the target population. Some exercises had to be entirely adapted since they consisted of poems, short stories, or informational paragraphs with high cultural relevance. Therefore, the original program development team instructed the research team to find information or stories relevant to the Portuguese culture that could interest people on topics that are not well known to most people in Portugal so they have something to learn and remember with each story. Table 2 presents the cultural adjustments made to the UCLA program during its adaptation to Portugal and the rationale behind such adaptations.

In week 1, the first exercises are auditory, and participants must count the number of occurrences of a given letter/pair of letters or word/words. The letter spelling records (15 files of letter spelling) did not need written translation because English and Portuguese use written systems based on the Latin alphabet. Therefore, the letter spelling files required only recording the spelling of those letters in PT-EU. Nevertheless, after the appraisal by the expert panel and target population, specific questions were raised about the alphabet. The alphabet used in the original program included the letters “K”, “W”, and “Y”. However, experts considered that not every person in the target population would be familiar with these letters, especially the older participants. This information was confirmed later with the target population. Therefore, following other Portuguese studies stating that, for most of the twentieth century, the letters “K”, “W”, and “Y” were not part of the alphabet taught in Portuguese primary schools [81], it was decided not to include those letters, even though current education programs include this set of letters and some foreign words written with these letters have since then entered the Portuguese lexicon. As such, those letters were randomly substituted by other letters.

The six auditory short stories records needed some adaptation (see Table 2); one story was only translated since it corresponded to general and not culture-related knowledge (“African Violets”). Major adaptations to the local culture concerned: (1) *food*—e.g., hotdog (PT-EU “cachorro-quente”) is more commonly found in Portuguese fairs than corndog; (2) *animals*—some animals mentioned in the texts are not commonly found in Portugal, such as pelicans, thus others, such as eels and lampreys, were added to fit the cultural context; (3) *unit of measurement*—“metros”, i.e., meters, to translate “feet”, a unit of measurement more common in the US; (4) and *museums, places or local information*—one of the original texts was about the Art Institute of Chicago, but since the cultural identification with this text by the Portuguese population would be little, it was replaced by a text about a similar Portuguese museum (Serralves Museum of Contemporary Art). Thus, considering the extent of adaptations made, one story was fully changed (“Art Institute of Chicago Story” to “Serralves Museum of Contemporary Art”), three stories were adapted with occasional cultural changes to specific words or expressions (“The County Fair Story”, “The Mechanic Story”, and “Maine Lighthouse Story”), and one had a few sentences adapted with information regarding the new cultural context (“The Ginkgo Tree Story”). With these adaptations, some exercise instructions needed changes (see Table 2).

The next exercises of week 1 were written exercises to practice visual attention/concentration, where participants are given matrices of letters and/or numbers, and they have to complete the blank spaces with the correct option that follows in a sequence. These exercises did not need to be adapted since English and Portuguese use the Latin alphabet.

Week 2 exercises were very similar to the written exercises of week 1 and thus did not need any adaptation.

Exercises of week 3 were the ones that suffered the most adaptations. Exercises related to training with checklists contained information essentially about food and customary practices. The two exercises of this type contained items (e.g., ingredients, utensils, organization practices, currency) that were not common in Portugal. Therefore, these were replaced by examples adapted to the local culture (see Table 2).

The next exercises concerned visual and auditory multitasking and consisted of both written and auditory content (i.e., poems, short stories, or informational paragraphs). Some (n = 6) of the visual multitasking exercises texts are similar to the Auditory Attention Exercises of week 1 (see Table 2). For these exercises, the instructions were adapted concerning the originals for only two of the texts, according to the new content; attention was given to words that were repeated a similar number of times, which would allow proposing similar exercises to those in the original program. For the other five texts, two were only translated with minor adaptations (e.g., an update of dates and population numbers). The only poem, “Tree, Tree” by the Spanish poet Federico García Lorca, was replaced by another similar poem from a Portuguese poet, “Adeus” by Eugénio de Andrade, after consulting a professor of Portuguese. The other two short stories were changed and replaced by other stories that maintained the original theme (“Canning” story to “Canned Food”, i.e., methods of food preservation; and “Volcanic Plugs” to “Volcanic Calderas”, i.e., geological structures, this one after consulting a professor of Geology). Concerning instructions, for three exercises, these were adapted according to the new or translated content, and the other two remained the same as the originals.

As for the auditory multitasking exercises, the three poems were changed, maintaining the genre, but the texts were replaced by poems originally written in Portuguese from Portuguese poets. For these exercises, instructions were changed according to the texts in a way that would allow the implementation of the exercises as in the original program (see Table 2). Three informational paragraphs were substituted (“The Great Zimbabwe”, “Mesa Verde”, and “Plato, the Story of a Cat”), keeping the genre (i.e., information about world history, travel destination with archaeological history, and traditional story) but including information that was more familiar to the local population. Although two of the texts were not initially adapted (namely “The Great Zimbabwe” and “Mesa Verde”), after the target population appraisal, it was decided that these texts needed to be about more familiar information to Portuguese people. The other five informational paragraphs were only translated, with some adaptations to the writing of the texts to keep them simple and without many English words to facilitate comprehension.

Finally, week 4 also has several exercises with references to cultural aspects (see Table 2). Food (e.g., sushi versus seafood), phone number, license plate, monuments (e.g., Washington monument versus Torre de Belém), street names, actor names (John Wayne versus Nicolau Breyner), and customary practices (e.g., baseball versus soccer) were some of the contents that needed adaptation to the Portuguese context.

Week 5 did not pose the need for any relevant adaptations.

### 3.2. Independent Review by a Panel of Experts

Most of the comments and suggestions that emerged from the qualitative appraisal performed by the experts were accepted without raising doubts. These comments referred to translations or phrasing, to improve comprehensibility. Some recommendations, especially those related to familiarity with terms, were identified and discussed later with the target population to reach a consensus. The expert panel also performed a quantitative appraisal for each session, covering six criteria (cf. Methods section), assessed on a scale of 1–4. Table 3 presents the median (Mdn) scores per criteria and session/global program.

None of the sessions nor criteria received scores of 1, indicating a full need for revision suggested by any expert. *Adequacy of names and other contents with a cultural character (criterion 5)* was considered “adequate, no need for revision” (Mdn = 4) for all sessions and the global program. Overall minor suggestions were accepted. This score was important to confirm that the contents subjected to cultural adaptation were well performed and adequate during the translation and adaptation step. Concerning the *sensitivity of the language used (criterion 1)*, only week 3 was rated as “adequate, in need of minor revisions” (Mdn = 3.5). There were suggestions to replace some terminology; for instance, “oncological survivors” (PT-EU “sobreviventes oncológicos”) was initially selected, but experts preferred the term “cancer survivors” (PT-EU “sobreviventes de cancro”) as it is more common. Even though some subjects of the target population will not identify themselves as “cancer survivors” [82], this terminology is widely used in the field of cancer [22]. Furthermore, “hormonal therapy” was preferred over “endocrine therapy” due to its familiarity, especially among breast cancer survivors. The scores given to *clarity and comprehension of contents (criterion 2)* ranged from “adequate, in need for minor revisions” (Mdn = 3 for weeks 2 and 4, or Mdn = 3.5 for week 1) and “adequate, no need for revision” (Mdn = 4, weeks 3 and 5). The qualitative appraisal was important to uniformize the way of treatment of participants throughout the manual (i.e., text inflected for the masculine gender and the third person of the singular) and to simplify/clarify/rewrite some sentences to facilitate the reading. *Familiarity and accessibility of expressions/terms/concepts (criterion 3)* had a similar appraisal by the experts’ panel. Three sessions were scored as “adequate, in need of minor revisions” (Mdn = 3 or Mdn = 3.5) and two with an Mdn = 4 “adequate, no need for revision”. Overall, minor suggestions were provided (e.g., the English word “preceded” has at least two translations in PT-EU “precedido” and “antecedido”, and both were suggested by different experts, with the consensus being reached only in the focus groups). There were also suggestions to clarify some *theoretical, technical and cultural concepts or terms* regarding their *precision (criterion 4)*, although the overall appraisal was good: three sessions rated as “adequate, no need for revision” (Mdn = 4) and two as “adequate, in need for minor revisions” (Mdn = 3.5). For instance, “control stress” was initially translated as “controlar o stress”, which was a more direct translation, but experts suggested that a more accurate translation would be “gerir o stress” (i.e., “manage stress”). The final criterion, *adequacy of cognitive exercises (criterion 6),* was the least well-rated criterion, with all sessions being rated as “adequate, in need of minor revisions” (Mdn = 3 or Mdn = 3.5). Several recommendations for change concerning instructions and presentation of the examples and respective exercises were made (see Section 3.1).

### 3.3. Appraisal by the Target Population

Eleven non-CNS cancer survivors participated in the focus group discussions. The majority of participants were female (72.73%), with a mean age of 45.55 years (SD = 12.2, 23–71 years), and all had completed high school (45.45%) or higher education (54.55%). Only three participants were not working at the time of the focus group (one student, one retired, and one unemployed). All females were diagnosed with breast cancer and males with hematological cancers. All participants had completed active treatments, with four participants still undergoing hormonal therapy.

Several terms were brought to the cancer survivors’ attention. These terms either raised doubts among the expert panel or the research team. One of the terms was “hormonal therapy”, which as with the panelists was preferred for being more familiar than “endocrine therapy”, the term originally used in the program. Another term that was discussed was “multitasking”. Most participants knew the concept and used it in their daily lives. However, some reported that the less literate population would probably not recognize the concept. Therefore, although its direct translation was also proposed (PT-EU “multitarefa”), it was preferred to maintain the technical term “multitasking” with an explanation between brackets (“performing multiple tasks simultaneously”). The alphabet without the letters “K”, “W”, and “Y” was also discussed, as previously mentioned (see Section 3.1). Another example was the word “checklist”. Even though it was a familiar term for most people, some did not know its meaning, especially older individuals. Therefore, it was suggested to use its Portuguese translation along with checklist “lista de verificação/checklist”, which was considered accessible for all.

The feedback from the target population was very important regarding the instructions and the content of some of the program’s exercises. For each session, an example of an exercise of every category was presented. In week 1, for the auditory exercises, it was suggested to specify the instruction “listen for and mark each occurrence of the word (…).” (PE-EU “faça uma marca na linha abaixo cada vez que ouvir a letra (…).”) by saying which mark they should use (e.g., a dash). It was also proposed to present an example before each exercise and apply the same strategy to every session. In week 3, the recipe and the bar-b-q were well-appraised as culturally adequate.

Regarding visual multitasking exercises, the adapted texts were well received, and it was indicated that for this exercise, the content of the texts was not an aspect that was crucial to perform the exercise. On the other hand, for auditory multitasking exercises, the content was considered important to complete the exercise. Having foreign words (e.g., in English) can pose an added difficulty to the exercise. Therefore, it was suggested to change some texts and choose texts more adapted to the European/Portuguese reality. Examples of these changes are the informational texts “The Great Zimbabwe” and “Mesa Verde”, which had difficult foreign words to understand (e.g., Karanga, Bantu, kopje, Anasazi).

Finally, focus group discussions were used to explore the target population’s perceived acceptability of the intervention components. Overall, all the participants considered the program appropriate and aligned with cancer survivors’ cognitive difficulties and underlying needs. In this context, three themes were identified that emerged from the thematic analysis: (i) perceived benefits, (ii) motivators and barriers to participation, and (iii) appraisal of the content and structure of the program.

Regarding benefits, participants consistently reported that the intervention could contribute to recognizing and minimizing the negative impact of treatments on cognitive function (e.g., memory, attention) by providing useful “tips” and “tools” for daily routine: *“(…) it’s important to know what the effects of chemobrain are because that, the fog, the loss of the ability to think quickly, the disorientation in traffic when there’s no reason because we take that route every day but get blocked in the middle of a roundabout”* (….) *there are strategies to combat it, and programs like this help and demystify it.”* [female, 40 years]. Some participants highlighted the positive effect of these approaches on returning to work after diagnosis in promoting greater acceptance and adaptation to new difficulties and demands. The potential contact with other cancer survivors was also reported as a benefit of participating in the intervention by enhancing the sharing and mutual validation of the difficulties and feelings experienced: *“The symptoms we experience are not just us who have them, there are other people who have them too, and it’s normal, so we don’t get so worried.”* [female, 41 years].

Among the main motivations for participating in a cognitive rehabilitation program, cancer survivors identified informational needs related to understanding the causal relationship between treatments and cognitive difficulties. A motivating factor for participation was also the perception of real difficulties at the memory level and the expectation of improvement through the intervention: “*I got very affected in terms of memory, essentially. I feel a really big difference. And I am motivated by the fact that perhaps I can improve (…), the exercises make me improve.”* [female, 43 years]. A major barrier anticipated to be an obstacle to participation was the time spent in the sessions and displacements and the consequent articulation with professional and family activities. For this reason, online interventions, identified as a preference for most participants, were perceived as an alternative to manage this limitation: “*I think that online becomes simpler because of traveling and time, and we can, perhaps, do as much online at home as I can do at work, at the end of the day. I think it becomes easier for that.”* [female, 43 years].

Regarding the appraisal of the content and structure of the program, all participants showed a high level of acceptability. Most survivors reported that the exercises included in the intervention proposal trained different cognitive abilities, were suitable for various audiences (e.g., different professional contexts), and presented an appropriate degree of difficulty: *“I think so. Because some of these exercises are for memorization, well, for memorization… concentration… of sequences” (…) I think that in general, these exercises are transversal to most professions, some apply on one side, others apply on the other.”* [female, 58 years]. However, for some participants, low levels of literacy and/or more limited intellectual and cognitive abilities were noted as potential barriers to performing exercises with greater complexity. The frequency and duration of the sessions (one session per week, for five weeks, for 2 h) was also considered adequate, especially if they were held in an online environment with greater schedule flexibility. All participants recognized the importance of doing the exercises outside the sessions to complement the intervention and showed openness to fitting them into their daily routines.

### 3.4. Systematization of Contributions, Improvement of the Final Text Materials of the Program, and Fidelity Check by the Original Program Development Team

Contributions that emerged from the expert panel and target population appraisals were finally systematized through meeting discussions among the Portuguese research team. Together, these contributions provided important inputs to improve the text materials of the program. These steps resulted in the *Draft 2* version of the program. All adaptations were reported in a template created by the research team, similar to Table 2, with the performed adaptations and their justification. This report was sent to the original program development team to archive and make a final check concerning the fidelity of the contents adapted since there was close contact between the teams during the adaptation process. The *Final Version* of the program was finally achieved. Future steps include implementing this version to test its acceptability, feasibility, and efficacy among Portuguese non-CNS cancer survivors.

## 4. Discussion

Research has found that culture plays an important role in the experience of cancer survivorship, influencing how people cope with cancer and how they respond, plan or act during survivorship [83]. Therefore, it is expected that interventions developed with culturally and linguistically adapted methods for one culture may not consistently fit the needs, values, or norms of another culture. With this in mind, this article describes the steps of the process to develop a culturally adapted version of the UCLA Cognitive Rehabilitation Intervention Program in Portugal-CanCOG^®^—Cognitive Rehabilitation in Cancer. As highlighted by initial data from the *needs assessment* (See Appendix A), several differences in sociodemographic characteristics between the UCLA program participants and potential Portuguese cancer survivors (some data from previous studies on group therapy conducted by the research team [84]) were observed. Therefore, this adaptation is crucial to implement the program in the new context successfully. Based on current guidelines, nine steps were identified and performed to guarantee the fidelity of the original program to a different culture.

Overall, this experience exemplifies the complexity of culturally adapting evidence-based interventions for international transference [59,61]. First, it is a time-consuming process. More than one year passed from the beginning of the content’s translation to the final version of the program. It also implies several meetings to discuss the findings and decide the best approach to follow. Furthermore, several participants are required to perform an accurate adaptation, with different backgrounds and sensitivity to the cultural nuances of a given country and culture, namely knowledge about cultural norms and familiarity with practices.

Changes were made both at the surface and deep structural level, considering not only surface-level translations but also deeper-level changes to reflect values and beliefs [49]. Many cultural adaptations require language translation or reading-level adjustments. Although it is considered a surface structure change, *translation* is a critical aspect of cultural adaptation. Additionally, culturally adapting an intervention goes beyond translating its materials to the target language. Direct translations of culture-specific idioms might not be possible, requiring challenging translations to preserve the original meaning [60]. Therefore, more than a direct translation of the materials, it was necessary to incorporate cultural expressions and messages that were culturally appropriate and meaningful and to use content that was reflective of the target population [85]. Thus, the collaboration of a professional translator was crucial during the translation of the UCLA program materials to maintain the fidelity of the adapted intervention program. The close involvement of the research team to *check the technical accuracy* of the translation and further *cultural adaptation* was also important, given their background in the main fields of the program and being natives of the target country.

Additionally, a culturally fit translation of the program was only possible thanks to the involvement of stakeholders, namely *experts* in cancer and neuropsychology and the *target population*. The integration of their perspectives to achieve a final version guarantees that the adapted program is concordant with the cultural habits and worldviews of the target population [42]. Still, the close contact and consultation with the original developers of the program and their *fidelity checks* throughout the adaptation process were crucial to receive advice on the original intent of the exercises and what could be suitable program adaptations [43].

Core components of the program were not altered, thus maintaining the fidelity of the essential elements related to the program theory [43]. Several adaptations were required; namely, words and expressions appropriate to the target population, considering reading- and education-level adjustments; words and expressions for semantic and conceptual equivalence; personal names and references to different cultural aspects, such as habits, customs, and traditions; and instructions and/or content of some exercises. The adapted materials are expected to help engage participants in the program; a culturally sensitive intervention may produce enhanced beneficial outcomes [40]. To this end, the future implementation of the program, as the last step of cultural adaptation, is crucial to assess the impact of the adapted intervention. It is expected that the program’s implementation will enable the demonstration of similar outcomes to those achieved by the original US version, demonstrating the efficacy of the Portuguese version of the UCLA program in improving cognitive complaints of cancer survivors.

Although cancer survivors reported positive feedback concerning the expected benefits of the program, potential challenges and barriers to a successful implementation of the intervention were also identified. Time spent in sessions, displacements, and consequent articulation with professional and family activities were pointed out as the main obstacles to participation. This is an important finding from this study since the literature is scarce regarding the experience and needs of cancer survivors with CRCI in cognitive rehabilitation intervention programs. An important qualitative study exploring the perceived facilitators and barriers to interventions found that ease of use, accessibility, and convenience are crucial to facilitate participation, while cost, time, intensiveness, and distance to travel hampered participation [86]. These results align with the preliminary findings of the current study concerning the potential participation in a cognitive rehabilitation program. In the Portuguese cancer setting, no data is available yet. Therefore, it is hoped that the findings from the needs assessment study will contribute to knowledge in this field that could help Portuguese researchers and clinicians to better plan the implementation of interventions to maximize their utilization and benefit. In an attempt to overcome displacement obstacles, the research team also intends to deliver the Portuguese version of the UCLA program in an online format (http://cancog.web.ua.pt/ accessed on 21 October 2022) [87], which was also reported by cancer survivors as an alternative to manage this limitation. Although the group format is perceived as beneficial due to the close contact with other cancer survivors (cf. [86]), providing an alternative online format could help reach other participants that otherwise would not participate in the intervention. This is also in line with previous international studies. For instance, van der Linden et al. [88] noticed that conventional in-person cognitive rehabilitation was not always accessible for every patient in clinical practice, considering its high demand and cost due to multiple hospital visits and lengthy face-to-face sessions with professionals. Therefore, to overcome some of these limitations, cognitive telerehabilitation was considered a valid possibility to increase the accessibility of the program in a cost-efficient mode of delivery.

This paper adds to the literature in several ways: (1) this is the first experience of adapting the UCLA program to a different language and cultural context and, therefore, it can provide guidance for other researchers that propose to do the same; (2) this is the first attempt, to the best of the authors knowledge, of adapting a cognitive rehabilitation program to a different country, based on a proposed model that aggregates different international guidelines with strategies from both bottom-up and top-down approaches, including multiple stakeholders; thus, this model can be replicated in the future to adapt other cognitive rehabilitation programs for other populations; (3) as this study demonstrates, successful cultural adaptation of a cognitive rehabilitation intervention is a planned, organized, iterative, and collaborative process [40]; it is hoped that the lessons learned throughout the adaption process of the UCLA program to Portugal provide guidance for other researchers and/or clinicians who will endeavor similar projects.

Finally, it should be noted that the adaptation of the UCLA program and development of a Portuguese version is highly relevant because it aligns with the aims of the European Consortium of Cancer and Cognition (ECCC), which states that post-cancer care cognitive intervention rehabilitation programs for CRCI should be promoted [10]. As their recent consensus paper indicates, actual specific actions in Europe are being developed, involved in the Innovative Partnership for Action Against Cancer (iPAAC) Joint Action initiative dedicated to public authorities, with a particular focus on cognitive intervention programs. One of the outcomes of the iPAAC’s Work Package 4—Integration in National Policies and Sustainability was a guide on practices and recommendations for the management of cognitive impairments after cancer (available at https://www.ipaac.eu/res/file/outputs/wp4/practices-recommendations-cancer-cognitive-impairements.pdf accessed on 21 October 2022), where the CanCOG^®^ program is indicated as one of the ongoing works to offer cognitive management in Europe. The study also highlights that CRCI should be a priority in multidisciplinary national and European cancer control programs. This is particularly important since CRCI has been under-addressed by oncology specialists, who are uncertain of potential management strategies [89,90], and cancer survivors receive, therefore, limited information about the possibility of occurrence of cognitive changes following cancer diagnosis and treatment [91,92].

## 5. Conclusions

This paper describes the process of cultural adaptation of the UCLA Cognitive Rehabilitation Intervention Program to Portugal. The main changes introduced to the original program materials are presented and discussed. A specific cultural adaptation model of nine steps for cognitive rehabilitation programs was proposed to adapt the UCLA program to the Portuguese context, which others can find useful if they attempt to undertake similar projects. Analysis of the main differences between the participants in the original program implementation and potential Portuguese participants was important to inform and guide the necessary changes; furthermore, the dynamic collaboration among project researchers and stakeholders, namely experts and the target population, was paramount to the implementation of these changes. Future studies will include implementing the original face-to-face group format and an equivalent individual web-based format to confirm their acceptability, feasibility, and efficacy in Portuguese cancer survivors. This type of program is an essential clinical tool to alleviate CRCI among cancer survivors and help improve their QoL. In Portuguese cancer-related settings, this program is a valuable contribution considering the absence of specific cognitive rehabilitation interventions for this population.

## Figures and Tables

**Figure 1 healthcare-11-00141-f001:**
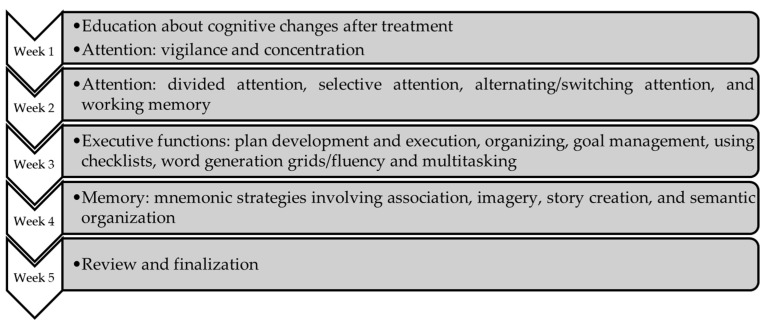
Overview of the UCLA Cognitive Rehabilitation Intervention Program contents and structure.

**Figure 2 healthcare-11-00141-f002:**
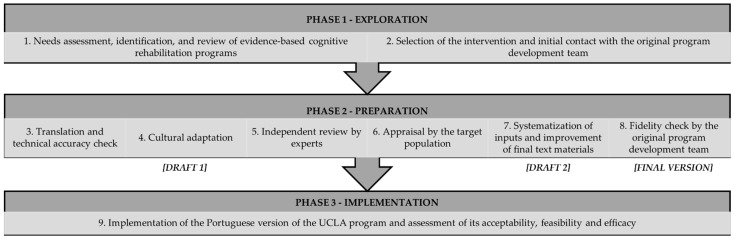
Framework for the cultural adaptation process of the UCLA Cognitive Rehabilitation Intervention Program to Portugal.

**Figure 3 healthcare-11-00141-f003:**
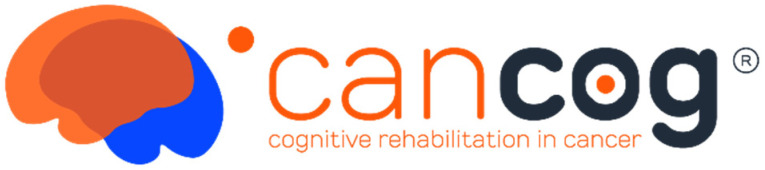
Name and logo of the Portuguese version of the UCLA program: CanCOG^®^—Cognitive Rehabilitation in Cancer.

**Table 1 healthcare-11-00141-t001:** UCLA Cognitive Rehabilitation Intervention Program.

Item of the TIDieR Checklist	Explanation
Brief Name	UCLA Cognitive Rehabilitation Intervention Program.
Why	The UCLA Cognitive Rehabilitation Intervention Program is a small group face-to-face cognitive rehabilitation intervention developed for the specific needs of cancer survivors with cognitive complaints. The program’s main aim is to improve self-reported and performance-based cognitive functioning.
What	The UCLA Cognitive Rehabilitation Intervention Program targets the most common complaints of cancer survivors (see Figure 1) and consists of three main components: (1) weekly education about cognitive functions at the beginning of each week related to the specific topic to be discussed (i.e., attention, executive functions, and memory) and training in technique instruction (e.g., deep breathing, muscle relaxation, countering negative thoughts); (2) in-class and homework cognitive exercises with three difficulty levels (level I—easy, level II—moderate and level III—difficult; levels I and II are performed in class to provide mastery experience, receive encouragement and support, engendering confidence and familiarity with the process; additional exercises of the three levels are assigned for homework); and (3) goal setting (short- and long-term goals, reviewed weekly and at the end of the intervention, respectively). Each of the components was derived from prior research on cognitive rehabilitation. Each session starts with a homework review (except week 1), followed by education, strategy training, in-class exercises, goal setting, and homework discussion. This intervention is a manualized program with teaching materials for the participants; the intervention manual was specifically developed for the program. There are four materials: Training Manual, Class Handouts, Home Practice Exercises, and Answer Keys, including audio files for cognitive exercises. Each participant receives a workbook with homework exercises, CDs for auditory exercises, answer keys, and a stopwatch for timing when needed. The participants are assigned homework relevant to their daily lives to make it easier to face challenges at home or work using strategies. Likewise, they are assigned exercises not discussed in class and are asked to set goals concerning specific daily life tasks that they need to accomplish. For the homework sessions, the participants are asked to track their homework practice on a log provided for them, explaining the importance of self-paced and distributed practice, and suggesting attempting at least four 20-min sessions per week of homework. The intervention is delivered by trained clinicians who use the Training Manual to guide them during the sessions. The manual includes references to handouts, which are printed and given to each participant, where they can give their answers to the exercises performed in class.
Who Provided	Intervention clinicians deliver the program. Clinicians are trained in the intervention content and monitored for the fidelity of delivery.
How	The intervention consists of five weekly sessions over five consecutive weeks. The intervention is delivered in a face-to-face small group format (3–9 participants) by trained clinicians.
Where	Face-to-face sessions take place at health or research facilities.
When and How Much	The intervention consists of five sessions over five consecutive weeks. Each session has 2 h in duration. Each component of the program has an approximate duration to be completed.
How Well	The feasibility study of the intervention showed promising results, being well-received and well-tolerated based on participants’ comments, their regular attendance, and the low attrition rate [27]. A randomized clinical trial evaluating the efficacy of the intervention compared with a waitlist control condition on cognitive complaints, and neuropsychological and brain functioning in breast cancer survivors was also conducted. Participants included in the intervention group showed immediate and sustained improvements in self-reported cognitive complaints and memory functioning on neurocognitive testing. These data were supported by the results of a quantitative electroencephalography (qEEG) substudy, providing some support for neurophysiological changes underlying the improvements in function associated with the intervention [28].

**Table 2 healthcare-11-00141-t002:** Cultural adjustments made to the UCLA program during its adaptation to Portugal and the rationale behind such adaptations.

Content of the Original Program	Cultural Adaptation	Reasons for Adaptation and Content Adapted
**WEEK 1—ATTENTION**		
Auditory Attention Exercises (Home Practice Exercises)		
1. “The County Fair Story”—instruction “Listen for and mark each occurrence of the word “they”.”	“The County Fair Story” [PT-EU: “A História da Feira Popular”]—instruction “Listen for and mark each occurrence of the word “in” (PT-EU “na”).”.	With the translation, the word “they” does not exist throughout the text. Since the word “they” is a pronoun, a word that was repeated the same number of times and that was similar in difficulty to a pronoun (i.e., a preposition) was found. Theme and plot are maintained, while the following elements were adapted to the local culture: “Feira Popular”, i.e., popular or people’s fair to translate “County Fair”. “Quinta dos animais domésticos”, i.e., domestic animals farm to translate “petting zoo”. “cabrito”, i.e., a baby goat or a kid, to translate “deer”, which is not a domestic animal. “Carrossel”, i.e., a carousel or a merry-go-round, to translate “Tilt-a-Whirl”, which consists of a number of cars that are “free to pivot about the center of its own circular platform” while “moving along a hilly circular that tilts the platform in all possible directions” [80]. “Ficar cheio de fome”, i.e., be starving or be hungry, to translate “work up an appetite”, which is less intense and is not frequently used in Portugal to justify the need to eat. “cachorro-quente”, i.e., hotdog, to translate “corndog”. “as pombas andaram […] a apanhar as migalhas que caíam‘’, i.e., “the pigeons were pecking the falling bread crumbs” to translate “pigeons begged at their feet” because “begging” is not a verb used in Portuguese to express such an idea.
2. “The Mechanic Story”	“The Mechanic Story” [PT-EU: “A História do Mecânico”]. Translated, with minor adaptations.	Theme and plot are maintained while the following element was adapted to the European unit of measurement: “metros”, i.e., meters, to translate “feet”, a unit of measurement more common in the US.
3. “Art Institute of Chicago Story”—instruction “Listen for and mark each occurrence of the word “Chicago”.”	“Serralves Museum of Contemporary Art” [PT-EU: “O Museu de Arte Contemporânea de Serralves”]. “Listen for and mark each occurrence of the word “Serralves”.”.	The text was changed by a text about a similar Portuguese museum to improve the cultural identification by the Portuguese population. The theme is maintained (i.e., a contemporary art museum), but because the object of the informative text is changed, all the information transmitted to the listener also changes.
4. “Maine Lighthouse Story”—instruction “Listen for and mark each occurrence of the word “would”.”	“Barra Lighthouse Story” (PT-EU “A História do Farol da Barra”)—instruction “Listen for and mark each occurrence of the word “lighthouse keeper” (PT-EU “faroleiro”).”	In PT-EU, the word “would” does not exist. Since the word “would” is a modal verb, a word that was repeated the same number of times and that was similar in difficulty to a verb (i.e., a noun) was found. The theme and plot are maintained, while the following elements were adapted to the local culture, language and characteristics of the object being described: “Farol da Barra”, i.e., the lighthouse of Barra (in Portugal), instead of Maine lighthouse, to localize the reference. “130 anos”, i.e., 130 years, instead of 120 years, for historic correctness. “famoso por ser o maior farol de Portugal”, i.e., famous for being the largest lighthouse in Portugal, instead of “famous for its woodwork and brass”, to maintain consistency of the information given on the object described. “jornalistas”, i.e., journalists, to translate “magazine writers”. “As gaivotas”, i.e., using only seagulls instead of “Seagulls and pelicans” because the second type of bird is not frequently found in the lighthouse of Barra or in Portugal and therefore is not commonly mentioned in daily language. “enguias, lampreias e caranguejos”, i.e., eels, lampreys and crabs, instead of “lobsters and crabs” because the first group of fishes and seafood is commonly consumed in several parts of Portugal.
5. “African Violets”	“African Violets” [PT-EU: “A História das Violetas-Africanas”]. Translated.	Translated.
6. “The Ginkgo Tree Story”	“The Ginkgo Tree Story” [PT-EU: “A História da Árvore Ginkgo”]. Translated, with minor adaptations.	The part of the text that referred to the US was adapted to Europe/Portugal.
**WEEK 3—EXECUTIVE FUNCTIONS**		
Checklist exercise: “Easy Sugar Cookie Recipe” (Training Manual)	“Easy Lemon Cookie Recipe”	The recipe for cookies was replaced by a very common recipe that was found online. The ingredients and utensils of the recipe were changed to best fit Portuguese customs.
Checklist exercise: “Plan a Pot-Luck Bar-B-Q” (Training Manual and Home Practice Exercises)	Translated, with major adaptations.	Although bar-b-qs are a very common practice in Portugal, their organization is somewhat different. First, the budget and the food prices were changed from American dollars to euros. Then, the main dish, appetizers, side dishes, beverages, and desserts were changed to best fit Portuguese customs.
Visual Multitasking Exercises (Home Practice Exercises)		
1. “The County Fair”—instructions “Circle words that are numbers (e.g., one or 1); Underline words that are food or drinks; Cross off the names of Fair rides.”.	Same as week 1	Same as week 1. Instructions are the same as the originals.
2. “The Art Institute of Chicago”—instructions “Underline “Chicago”; Cross-out the word “art”.”	Same as week 1. Instructions “Underline “Serralves”; Cross-out the word “art”.”	Same as week 1. Instructions were adapted accordingly, substituting “Chicago” by “Serralves”.
3. “Poem Tree, Tree by Federico García Lorca”—instructions “Circle colors; Underline “girl”.”	“Poem Adeus by Eugénio de Andrade”—instructions “Circle body parts; Underline “clouds”.”	The genre is maintained as in the original program but the text is replaced by a poem originally written in Portuguese. The main objective was to use a poem that would allow proposing and replicating similar exercises to those in the original program (i.e., to circle/underline/cross off specific words). Instructions were adapted accordingly.
4. “The Mechanic”—instructions “Circle numbers; Underline tools; Cross-out types of clothing.”	Same as week 1.	Same as week 1. Instructions are the same as the originals.
5. “Geography of South America”—instructions “Circle types of animals; Underline numbers; Cross-out types of water formations.”	Translated, with minor adaptations.	Theme and information given were maintained while the following element was updated: Estimated population in South America from 371 million in 2005 to 423 “milhões” in 2018. Instructions are the same as the originals.
6. “Canning”—instructions “Circle the word “lid”; Underline the word “canning”; Cross-out types of food.”	“Canned Food” [PT-EU “Comida em Conserva ou Enlatada”]—instructions “Circle the word “can” (PT-EU “latas”; Underline the word “canned” (PT-EU “enlatada”); Cross-out types of food.”	The text was initially translated, but it was later changed since home canning, as occurs in the US, is not common in Portugal (i.e., several processes are unfamiliar to the Portuguese population). Therefore, it was replaced by canned food, which is similar to home canning. Canned food is very common in Portugal. Instructions were adapted accordingly.
7. “African Violets”—instructions “Circle the word “violet” (or “violets”); Underline colors; Cross-out numbers.”	Same as week 1.	Same as week 1. Instructions are the same as the originals.
8. “The Ginkgo Tree”—instructions “Circle the word “Ginkgo” (or “Ginkgos”); Underline parts of plants; Cross-out the word “and”.”	Same as week 1.	Same as week 1. Instructions are the same as the originals.
9. “The Maine Lighthouse”—instructions “Circle professions; Underline animals; Cross-out words that have to do with time.”	“The Barra Lighthouse” (PT-EU “O Farol da Barra”)—instructions “Circle professions; Underline animals; Cross-out words that have to do with time.”	The text was changed to a Portuguese lighthouse located in Aveiro, the Barra Lighthouse. The text was only adapted when it referred to information about Maine and animals. Instructions were adapted accordingly.
10. “Zebras and Zebra Stripes”—instructions “Circle the word “stripes” (or “stripe”); Cross-out the word “Zebras” (or “Zebra”).”	Translated.	Instructions are the same as the originals.
11. “Volcanic Plugs”—instructions “Circle the names of countries or states; Underline the word plug (or plugs); Cross-out the word “the”.”	“Volcanic Calderas” [PT-EU: “Caldeiras Vulcânicas”]—instructions “Circle the names of continents, countries or cities/islands; Underline the word “caldera” (PT-EU “caldeira”) (or “calderas”; PT-EU “caldeiras”); Cross-out the word “the”.”	The original text, when translated into PT-EU, raised many doubts because several technical terms did not have a specific Portuguese translation. After consulting a professor of Geology, it was decided to change that text for one with a similar theme which included examples from Portugal (Azores). Instructions were adapted accordingly.
Auditory Multitasking Exercises (Home Practice Exercises)		
1. Sunset (Poem by Lawrence Dunbar)	A Dor que a Minha Alma Sente (Poem by Luís Vaz de Camões)	The genre is maintained but the texts are replaced by poems originally written in Portuguese. Rather than maintaining the theme, the main objective was to use poems that would allow the implementation of the exercises in the original program (i.e., count repeated or specific words and listen for a given word and report the text that follows). Instructions were also changed according to the texts, but the research team tried to keep as faithful as possible to the original.
2. Ozymandius of Egypt (Poem by Percy Shelley)	Poema (Poem by Sophia de Mello Breyner Andresen)	The genre is maintained but the texts are replaced by poems originally written in Portuguese. Rather than maintaining the theme, the main objective was to use poems that would allow the implementation of the exercises in the original program (i.e., count repeated or specific words and listen for a given word and report the text that follows). Instructions were also changed according to the texts, but the research team tried to keep as faithful as possible to the original.
3. The Tiger (Poem by William Blake)	Tenho Tanto Sentimento (Poem by Fernando Pessoa)	The genre is maintained but the texts are replaced by poems originally written in Portuguese. Rather than maintaining the theme, the main objective was to use poems that would allow the implementation of the exercises in the original program (i.e., count repeated or specific words and listen for a given word and report the text that follows). Instructions were also changed according to the texts, but the research team tried to keep as faithful as possible to the original.
4. The Great Zimbabwe (informational paragraph)	The Rome Coliseum (informational paragraph)	The genre is maintained (i.e., information about world history) but replaced by informational text about a European place.
5. Mesa Verde (informational paragraph)	Ruins of Conímbriga (informational paragraph)	The genre is maintained (i.e., information about travel destination with archaeological history) but replaced by analogous informational text about a Portuguese place (i.e., an archaeological place in Portugal).
6. Plato, the Story of a Cat (Story by A.S. Downs)	O Pastor feito Mercador (short story of Portuguese folklore by an unknown author)	The genre is maintained (i.e., traditional story) but replaced by a traditional short story of the Portuguese folklore that would allow implementing the exercises in the original program (i.e., questions about the content of the story).
7. Ice on Mars (informational paragraph)	Ice on Mars (informational paragraph)	Translated. The research team tried to keep the texts simple and without English words to facilitate comprehension.
8. Zebras and Zebra Stripes (informational paragraph)	Zebras and Zebra Stripes (informational paragraph)	Translated. The research team tried to keep the texts simple and without English words to facilitate comprehension.
9. Kaleidoscopes	Kaleidoscopes	Translated. The research team tried to keep the texts simple and without English words to facilitate comprehension. Some instructions were also adapted (e.g., two questions referred to information regarding two personalities with foreign names, but since the names were in English, these questions were adapted).
10. Igloos	Igloos	Translated. The research team tried to keep the texts simple and without English words to facilitate comprehension.
11. Invention of the Wheel	Invention of the Wheel	Translated. The research team tried to keep the texts simple and without English words to facilitate comprehension.
**WEEK 4—MEMORY**		
License plate number is: ZOD336 (Training Manual)	License plate number is: CA 59 LP	The license plate number was changed for the Portuguese context.
Get a key made for your long-lost cousin who will visit (Training Manual)	Get a key made for the younger son that goes to school	Getting a key for a cousin is not a Portuguese habit.
Make reservations for Sushi (Training Manual)	Make reservations for seafood restaurant	Although sushi is becoming more common in Portugal, it is more common to go eat shellfish (especially among 40–65 years).
Brunch (Training Manual)	Lunch	Brunch is also very common only for younger generations.
“Snakebite Junction” by Harry Kissinger (Home Practice Exercises)	Mar Me Quer by Mia Couto	The name was changed to a Portuguese name.
Crocker Canyon Exit off the 52 East Freeway (Home Practice Exercises)	Saída para Paço de Arcos da A5 em direção a Cascais	The name of the streets was adapted.
Chinese Restaurant called Ming How (Home Practice Exercises)	A Brazilian Restaurant called Estrela do Sul (Star of the South)	Although Chinese restaurants are becoming more familiar in Portugal, some people do not go to or know Chinese food.
Beecher Ave. (Home Practice Exercises)	Avenida de Ceuta	The name of the street was adapted.
Phone number 368-7621 (Home Practice Exercises)	Phone number 234370200	The phone number was adapted.
Baseball mitt (Home Practice Exercises)	Soccer gloves	Soccer is more common in Portugal.
John Wayne (Home Practice Exercises)	Nicolau Breyner	Actor adapted.
Marshmallow (Home Practice Exercises)	Candy	Marshmallow is not very common in Portugal.
Washington Monument (Home Practice Exercises)	Torre de Belém	Monument in the country’s capital.

Color and bold highlight the structure of the program.

**Table 3 healthcare-11-00141-t003:** Expert panel quantitative appraisal for each session/global program of the UCLA program concerning six criteria.

Week	1. Sensitiveness (Mdn)	2. Clarity and Comprehensibility (Mdn)	3. Familiarity and Accessibility (Mdn)	4. Precision (Mdn)	5. Cultural Adequacy (Mdn)	6. Adequacy of Cognitive Exercises (Mdn)
#1	4	3.5	3.5	4	4	3.5
#2	4	3	3.5	4	4	3
#3	3.5	4	4	3.5	4	3
#4	4	3	3	3.5	4	3
#5	4	4	4	4	4	3
Global program	4	3	4	3	4	3

## Data Availability

The data that support the findings of this study are available from the corresponding authors upon reasonable request.

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
