# Peer review of "CanCOG®: Cultural Adaptation of the Evidence-Based UCLA Cognitive Rehabilitation Intervention Program for Cancer Survivors in Portugal"

_healthcare, 2023, doi:10.3390/healthcare11010141_

Round 1
Reviewer 1 Report
Overal a very detailed manuscript describing the cultural adaptation of a cancer rehab program for the use in Portugal. It is clearly written and very detailed. This should be helpful if anyone else intends to perform the same type of work.
A few comments:
Line 957: …adapt the F&S! program - what is meant???
Lines 968-973: What is meant here???
The manuscript also needs to be linguistically checked. For example, data is... should be data are... Normally we write (depending on the context) "in the US".
Reviewer 2 Report
Thank you for allowing me to review. Please find the attached comments and edits on word document.
PDF with comments sent to editors
